# Polish evaluation dataset
# for compositional distributional semantics models

## Abstract

The paper presents a procedure of building an evaluation dataset for validation of compositional distributional semantics models estimated for languages other than English. The procedure generally builds on steps designed to assemble the SICK corpus which contains pairs of English sentences annotated for semantic relatedness and entailment, as we aim at building a comparable dataset. However, the implementation of particular building steps significantly differs from the original SICK design assumptions, which is caused by both lack of necessary extraneous resources for an investigated language and the demand for language-specific transformation rules. The designed procedure is verified on Polish, a fusional language with a relatively free word order, and contributes to building a Polish evaluation dataset. The resource consists of 10K sentence pairs which are human-annotated for semantic relatedness and entailment. The dataset may be used for evaluation of compositional distributional semantics models of Polish.

## 1 Introduction and related work

Can you imagine a straightforward answer to the question *How to automatically analyse semantics of a natural language and to represent the meaning of phrases (or even sentences) in this language in an accessible way?* A few years ago most of us would probably answer *I don't know!* And now, in the era of high-speed and high-performance computing, everybody seems to *know it* – with distributional semantics models.

### 1.1 Distributional semantics

The basic idea of distributional semantics, i.e. determining the meaning of a word based on its co-occurrence with other words, is derived from the empiricists – Harris (1954) and Firth (1957). John R. Firth drew attention to the context-dependent nature of meaning especially with his famous maxim "You shall know a word by the company it keeps" (Firth, 1957, p. 11).

Nowadays, distributional semantics models are estimated with various methods, e.g. word embedding techniques (Bengio et al., 2003, 2006; Mikolov et al., 2013). To ascertain the purport of a word, e.g. *bath*, you can use the context of other words that surround it. If we assume that the meaning of this word expressed by its lexical context is associated with a distributional vector, the distance between distributional vectors of two semantically similar words, e.g *bath* and *shower*, should be smaller than between vectors representing semantically distinct words, e.g. *bath* and *tree*.

### 1.2 Compositional distributional semantics

Based on empirical observations that distributional vectors encode certain aspects of the word meaning, it is expected that similar aspects of the meaning of phrases and sentences can also be represented with vectors obtained via composition of distributional word vectors. The idea of semantic composition is not new. It is well known as *principle of compositionality*:[1] "The meaning of a compound expression is a function of the meaning of its parts and of the way they are syntactically combined." (Janssen, 2012, p. 19).

Modelling the meaning of textual units larger than words using compositional and distributional information is the main subject of compositional

---

[1] As the principle of compositionality is attributed to Gottlob Frege, it is often called *Frege's principle*.

distributional semantics (Mitchell and Lapata, 2010; Baroni and Zamparelli, 2010; Grefenstette and Sadrzadeh, 2011; Socher et al., 2012, to name a few studies). Fundamental principles of compositional distributional semantics, henceforth referred to as CDS, are mainly propagated with papers written on the topic. Apart from papers it was the SemEval-2014 Shared Task 1 (Marelli et al., 2014) which essentially contributed to the expansion of CDS and increased an interest in this domain. The goal of the task was to evaluate CDS models of English in terms of semantic relatedness and entailment on proper sentences from the SICK corpus.

### 1.3 The SICK corpus

The SICK corpus (Bentivogli et al., 2014) consists of 10K pairs of English sentences containing multiple lexical, syntactic, and semantic phenomena. It builds on two external data sources – the 8K ImageFlickr dataset (Rashtchian et al., 2010) and SemEval-2012 Semantic Textual Similarity dataset (Agirre et al., 2012). Each sentence pair is human-annotated for relatedness in meaning and entailment.

The relatedness score corresponds to the degree of semantic relatedness between two sentences and is calculated as the average of ten human ratings collected for this sentence pair on the 5-point Likert scale. This score indicates the extent to which the meanings of two sentences are related.

The entailment relation between two sentences, in turn, is labelled with *entailment*, *contradiction*, or *neutral*. According to the SICK guidelines, the label assigned by the majority of human annotators is selected as the valid entailment label.

### 1.4 Motivation and organisation of the paper

Studying approaches to various natural language processing (henceforth NLP) problems, we have observed that the availability of language resources (e.g. training or testing data) stimulates development of NLP tools and estimation of NLP models. Hence, we aim at building datasets for evaluation of CDS models in languages other than English, which are often under-resourced. We therefore assume that availability of test data will encourage development of CDS models in these languages.

We start with a high-quality dataset for Polish, which is a completely different language than English in at least two dimensions. First, it is

a rather under-resourced language in contrast to the resource-rich English. Second, it is a fusional language with a relatively free word order in contrast to the isolated English with a relatively fixed word order.

The procedure of building an evaluation dataset for validating compositional distributional semantics models of Polish generally builds on steps designed to assemble the SICK corpus (described in Section 1.3), as we aim at building an evaluation dataset which is comparable to the SICK corpus. However, the implementation of particular building steps significantly differs from the original SICK design assumptions, which is caused by both lack of necessary extraneous resources for Polish (see Section 2.1) and demand for Polish-specific transformation rules (see Section 2.2). Furthermore, rules of arranging sentences into pairs (see Section 2.3) are defined anew taking into account the characteristic of data and bidirectional entailment annotations, since an entailment relation between two sentences must not be symmetric. Even if our assumptions of annotating sentence pairs coincide with the SICK principles to a certain extent (see Section 3.1), the annotation process differs from the SICK procedure, in particular by introducing an element of human verification of correctness of automatically transformed sentences (see Section 3.2) and some additional post-corrections (see Section 3.3). Finally, a summary of the dataset is provided in Section 4.1 and the dataset evaluation is given in Section 4.2.

## 2 Procedure of collecting data

### 2.1 Selection and description of images

The first step of building the SICK corpus consisted in the random selection of English sentence pairs from existing datasets (Rashtchian et al., 2010; Agirre et al., 2012). Since we are not aware of accessibility of analogous resources for Polish, we have to select images first and then describe the selected images.

Images are selected from the 8K ImageFlickr dataset (Rashtchian et al., 2010). At first we wanted to take only these images the descriptions of which were selected for the SICK corpus. However, a cursory check shows that these images are quite homogeneous, with a predominant number of dogs depictions. Therefore, we independently extract 1K images and split them into 46 thematic groups (e.g. *children*, *musical instruments*, *motor-*

*bikes*, *football*, *dogs*). Numbers of images within individual thematic groups vary from 6 images in the *volleyball* and *telephoning* groups to 94 images in the *various people* group. The second largest groups are *children* and *dogs* with 50 images each.

The chosen images are given to two authors who independently of each other formulate their descriptions based on a short instruction. The authors are instructed to write one single sentence (with a sentence predicate) describing the action on a displayed image. They should not describe an imaginable context or an interpretation of what may lie behind the scene on the picture. If some details on the picture are not obvious, they should not be described too. Furthermore, the authors should avoid multiword expressions, such as idioms, metaphors, and named entities, as those are not compositional linguistic phenomena. Finally, descriptions should contain Polish diacritics and proper punctuation.

## 2.2 Transformation of descriptions

The second step of building the SICK corpus consisted in pre-processing extracted sentences, i.e. normalisation and expansion (Bentivogli et al., 2014, p. 3–4). Since the authors of Polish descriptions are asked to follow the guidelines (presented in Section 2.1), the normalisation step is not essential for our data. The expansion step, in turn, is implemented and the sentences provided by the authors are lexically and syntactically transformed in order to obtain derivative sentences with similar, contrastive, or neutral meanings. The following transformations are implemented:

1. *dropping conjunction* concerns sentences with coordinated predicates sharing a subject, e.g. *Rowerzysta odpoczywa i obserwuje morze.* (Eng. 'A cyclist is resting and watching the sea.'). The finite form of one of the coordinated predicates is transformed into:

   - an active adjectival participle, e.g. *Odpoczywający rowerzysta obserwuje morze.* (Eng. 'A resting cyclist is watching the sea.') or *Obserwujący morze rowerzysta odpoczywa.* (Eng. 'A cyclist, who is watching the sea, is resting.'),
   - a contemporary adverbial participle, e.g. *Rowerzysta, odpoczywając, obserwuje morze.* (Eng. 'A cyclist is watching the sea, while resting.') or *Rowerzysta odpoczywa, obserwując morze.*

(Eng. 'A cyclist is resting, while watching the sea.').

2. *removing conjunct in adjuncts*, i.e. the deletion of one of coordinated elements of an adjunct, e.g. *Mały, ale zwinny kot miauczy.* (Eng. 'A small but agile cat miaows.') can be changed into either *Mały kot miauczy.* (Eng. 'A small cat miaows.') or *Zwinny kot miauczy.* (Eng. 'An agile cat miaows.').

3. *passivisation*, e.g. *Człowiek ujeżdża byka.* (Eng. 'A man is breaking a bull in.') can be transformed into *Byk jest ujeżdżany przez człowieka.* (Eng. 'A bull is being broken in by a man.').

4. *removing adjuncts*, e.g. *Dwa białe króliki siedzą na trawie.* (Eng. 'Two small rabbits are sitting on the grass.') can be changed into *Króliki siedzą.* (Eng. 'The rabbits are sitting.').

5. *swapping relative clause for participles*, i.e. a relative clause swaps with a participle (and vice versa), e.g. *Kobieta przytula psa, którego trzyma na smyczy.* (Eng. 'A woman hugs a dog which she keeps on a leash.'). The relative clause is interchanged for a participle construction, e.g. *Kobieta przytula trzymanego na smyczy psa.* (Eng. 'A woman hugs a dog kept on a leash.').

6. *negation*, e.g. *Mężczyźni w turbanach na głowie siedzą na słoniach.* (Eng. 'Men in turbans on their heads are sitting on elephants.') can be transformed into *Nikt nie siedzi na słoniach.* (Eng. 'Nobody is sitting on elephants.'), *Żadni mężczyźni w turbanach na głowie nie siedzą na słoniach.* (Eng. 'No men in turbans on their heads are sitting on elephants.'), and *Mężczyźni w turbanach na głowie nie siedzą na słoniach.* (Eng. 'Men in turbans on their heads are not sitting on elephants.').

7. *constrained mixing of dependents from various sentences*, e.g. *Dwoje dzieci siedzi na wielbłądach w pobliżu wysokich gór.* (Eng. 'Two children are sitting on camels near high mountains.') can be changed into *Dwoje dzieci siedzi przy zastawionym stole w pobliżu wysokich gór.* (Eng. 'Two children

are sitting at the table laden with food near high mountains.').

The first five transformations are designed to produce sentences with a similar meaning, the sixth transformation outputs sentences with a contradictory meaning, and the seventh transformation should generate sentences with a neutral (or unrelated) meaning. All transformations are performed on dependency structures of input sentences (Wróblewska, 2014).

Some of the transformations are very productive (e.g. mixing dependents). Other, in turn, are sparsely represented in the output (e.g. dropping conjunction). The number of transformed sentences randomly selected to build the dataset is in the second column of Table 1.

| transformation | selected | |
| --- | --- | --- |
| dropping conjunction | 139 | 2.0% |
| removing conjunct in adjunct | 485 | 6.9% |
| passivisation | 893 | 12.8% |
| removing adjuncts | 1013 | 14.5% |
| swapping rc↔ptcp | 1291 | 18.4% |
| negation | 1304 | 18.6% |
| mixing dependents | 1878 | 26.8% |

Table 1: Numbers of transformed sentences selected for annotation.

### 2.3 Data ensemble

The final step of building the SICK corpus consisted in arranging normalised and expanded sentences into pairs. As our data diverges from SICK data, the process of arranging Polish sentences into pairs also differs from pairing in the SICK corpus. Apart from pairs connecting two sentences originally written by humans (as described in Section 2.1), there are also pairs in which an original sentence is connected with a transformed sentence. For each of the 1K images, the following 10 pairs are constructed (for $A$ being the set of all sentences originally written by the first author, $B$ being the set of all sentences originally written by the second author, $\mathbf{a} \in A$ and $\mathbf{b} \in B$ being the original descriptions of the picture):

1. $(\mathbf{a}, \mathbf{b})$,

2. $(\mathbf{a}, \mathbf{a}_1)$, where $\mathbf{a}_1 \in t(\mathbf{a})$, for $t(\mathbf{a})$ being the set of all transformations of the sentence $\mathbf{a}$,

3. $(\mathbf{b}, \mathbf{b}_1)$, where $\mathbf{b}_1 \in t(\mathbf{b})$,

4. $(\mathbf{a}, \mathbf{b}_2)$, where $\mathbf{b}_2 \in t(\mathbf{b})$,

5. $(\mathbf{b}, \mathbf{a}_2)$, where $\mathbf{a}_2 \in t(\mathbf{a})$,

6. $(\mathbf{a}, \mathbf{a}_3)$, where $\mathbf{a}_3 \in t(\mathbf{a}'), \mathbf{a}' \in A, \mathcal{T}(\mathbf{a}') = \mathcal{T}(\mathbf{a}), \mathbf{a}' \neq \mathbf{a}$, for $\mathcal{T}(\mathbf{a})$ being the thematic group of $\mathbf{a}$,

7. $(\mathbf{b}, \mathbf{b}_3)$, where $\mathbf{b}_3 \in t(\mathbf{b}'), \mathbf{b}' \in B, \mathcal{T}(\mathbf{b}') = \mathcal{T}(\mathbf{b}), \mathbf{b}' \neq \mathbf{b}$,

8. $(\mathbf{a}, \mathbf{a}_4)$, where $\mathbf{a}_4 \in A, \mathcal{T}(\mathbf{a_4}) \neq \mathcal{T}(\mathbf{a})$,

9. $(\mathbf{b}, \mathbf{b}_4)$, where $\mathbf{b_4} \in B, \mathcal{T}(\mathbf{b_4}) \neq \mathcal{T}(\mathbf{b})$,

10. $(\mathbf{a}, \mathbf{a}_5)$, where $\mathbf{a}_5 \in t(\mathbf{a}), \mathbf{a}_5 \neq \mathbf{a}_1$ for 50% images, $(\mathbf{b}, \mathbf{b}_5)$ (analogously) for other 50%.

For each sentence pair $(\mathbf{a}, \mathbf{b})$ created according to this procedure, its reverse $(\mathbf{b}, \mathbf{a})$ is also included in our corpus. As a result, the working set consists of 20K sentence pairs.

## 3 Corpus annotation

### 3.1 Annotation assumptions

The degree of semantic relatedness between two sentences is calculated as the average of all human ratings on the Likert scale with the range from 0 to 5. Since we do not want to excessively influence the annotations, the guidelines given to annotators are mainly example-based:

- 5 (very related): *Kot siedzi na płocie.* (Eng. 'A cat is sitting on the fence.') vs. *Na płocie jest duży kot.* (Eng. 'There is a large cat on the fence.'),

- 1–4 (more or less related):
  *Kot siedzi na płocie.* (Eng. 'A cat is sitting on the fence.') vs. *Kot nie siedzi na płocie.* (Eng. 'A cat is not sitting on the fence.');
  *Kot siedzi na płocie.* (Eng. 'A cat is sitting on the fence.') vs. *Właściciel dał kotu chrupki.* (Eng. 'The owner gave kibble to his cat.');
  *Kot siedzi na płocie.* (Eng. 'A cat is sitting on the fence.') vs. *Kot miauczy pod płotem.* (Eng. 'A cat miaows under the fence.').

- 0 (unrelated): *Kot siedzi na płocie.* (Eng. 'A cat is sitting on the fence.') vs. *Zaczął padać deszcz.* (Eng. 'It starts raining.').

Apart from these examples, there is a note in the annotation guidelines indicating that the degree of semantic relatedness is not the same thing as the degree of semantic similarity. Since semantic similarity is only a special case of semantic relatedness, semantic relatedness is thus a more general term than the other one.

Polish entailment labels correspond directly to the SICK labels (i.e. *entailment*, *contradiction*, *neutral*). The entailment label assigned by the majority of human judges is selected as the gold label. The entailment labels are defined as follows:

- a **wynika z** b (b entails a) – if a situation or an event described by the sentence b occurs, it is recognised that a situation or an event described by a occurs as well, i.e. a and b refer to the same event or the same situation,

- a **jest zaprzeczeniem** b (a is the negation of b) – if a situation or an event described by b occurs, it is recognised that a situation or an event described by a may not occur at the same time,

- a **jest neutralne wobec** b (a is neutral to b) – the truth of a situation described by a cannot be determined on the basis of b.

### 3.2 Annotation procedure

Similarly as in the SICK corpus, each Polish sentence pair is human annotated for semantic relatedness and entailment by 3 human judges experienced in Polish linguistics.[2] Since for each annotated pair (a, b), its reverse (b, a) is also subject to annotation, the entailment relation is in practice determined 'in both directions' for 10K sentence pairs. For the task of relatedness annotation, the order of sentences within pairs seems to be irrelevant, we can thus assume to obtain 6 relatedness scores for 10K unique pairs.

Since the transformation process is fully automatic and to a certain extent based on imperfect dependency parsing, we cannot ignore errors in the transformed sentences. In order to avoid annotating erroneous sentences, the annotation process is divided into two stages:

1. a sentence pair is sent to a judge with the *leader* role, who is expected to edit and

---

[2]Our annotators have relatively strong linguistic background. Five of them have PhD in linguistics, five are PhD students, one is a graduate, and one is an undergraduate.

to correct the transformed sentence from this pair before annotation, if necessary,

2. the verified and possibly enhanced sentence pair is sent to the other two judges, who can only annotate it.

The *leader* judges should correct incomprehensible and ungrammatical sentences with a minimal number of necessary changes. Unusual sentences which are generally accepted by Polish speakers should not be modified. Moreover, the modified sentence may not be identical with the other sentence in the pair.

### 3.3 Impromptu post-corrections

During the annotation process it came out that sentences accepted by some human annotators are unacceptable for other annotators. We thus decided to garner annotators' comments and suggestions for improving sentences. After validation of these suggestions by an experienced linguist, it turns out that most of these proposals concern punctuation errors and typos in 228 distinct sentences. These errors are fixed directly in the corpus, as they should not impact annotations of sentence pairs. The other suggestions concern more significant changes in 141 distinct sentences. Annotations of pairs with modified sentences are resent to the annotators so that they can verify and update them.

## 4 Corpus summary and evaluation

### 4.1 Corpus statistics

Tables 2 and 3 summarise the annotations of the resulting 10K sentence pairs corpus. Table 2 aggregates the occurrences of 6 possible relatedness scores, calculated as the mean of all 6 individual annotations, rounded to an integer.

| relatedness | # of pairs |
|---|---|
| 0 | 1977 |
| 1 | 1430 |
| 2 | 1081 |
| 3 | 2165 |
| 4 | 2384 |
| 5 | 963 |

Table 2: Final relatedness scores rounded to integers (total: 10K pairs).

Table 3 shows the number of the particular entailment labels in the corpus. Since each sentence

pair is annotated for entailment in both directions, the final entailment label is actually a pair of two labels:

- *entailment+neutral* points to 'one-way' entailment,

- *contradiction+neutral* points to 'one-way' contradiction,

- *entailment+entailment*, *contradiction+contradiction*, and *neutral+neutral* point to equivalence.

While the actual corpus labels are ordered in the sense that there is a difference between, e.g. *entailment+neutral* and *neutral+entailment* (the entailment occurs in different directions), we treat all labels as unordered for the purpose of this summary (e.g. *entailment+neutral* covers *neutral+entailment* as well, representing the same type of relation between two sentences). The *X* label corresponds to pairs for which a majority label could not be established.

| entailment | # of pairs |
|---|---|
| *neutral+neutral* | 6477 |
| *entailment+neutral* | 1741 |
| *entailment+entailment* | 932 |
| *contradiction+contradiction* | 717 |
| *contradiction+neutral* | 113 |
| *X* | 20 |

Table 3: Final entailment labels (total: 10K pairs).

## 4.2 Inter-annotator agreement

The standard measure of inter-annotator agreement in various natural language labelling tasks is Cohen's kappa (Cohen, 1960). However, this coefficient is designed to measure agreement between two annotators only. As there are three annotators of each pair of ordered sentences, we decided to apply Fleiss' kappa[3] (Fleiss, 1971) designed for measuring agreement between multiple raters who give categorical ratings to a fixed number of items. An additional advantage of this measure is that different items can be rated by different human judges, which doesn't impact measurement. The normalised Fleiss' measure of inter-annotator agreement is:

---

[3]As Fleiss' kappa is actually generalisation of Scott's $\pi$ (Scott, 1955), it is sometimes referred to as Fleiss' *multi-*$\pi$, cf. Artstein and Poesio (2008).

$$\kappa = \frac{\bar{P} - \bar{P}_e}{1 - \bar{P}_e}$$

where the quantity $\bar{P} - \bar{P}_e$ measures the degree of agreement actually attained in excess of chance, while "[t]he quantity $1 - \bar{P}_e$ measures the degree of agreement attainable over and above what would be predicted by chance" (Fleiss, 1971, p. 379).

We recognise Fleiss' kappa as particularly useful for measuring inter-annotator agreement with respect to entailment labelling in our evaluation dataset. First, there are more then two raters. Second, entailment labels are categorial. Measured with Fleiss' kappa, there is inter-annotator agreement of $\kappa = 0.732$ for entailment labels in Polish evaluation dataset, which is quite satisfactory as for a semantic labelling task.

Relative to semantic relatedness, the distinction in meaning of two sentences made by human judges is often very subtle. This is also reflected in the inter-annotator agreement scores measured with Fleiss' kappa. Inter-annotator agreement measured for six semantic relatedness groups corresponding to points on the Likert scale is quite low: $\kappa = 0.336$. If we measure inter-annotator agreement for three classes corresponding to the three relatedness groups from the annotation guidelines (see Section 3.1), i.e. <0>, <1, 2, 3, 4>, and <5>, the Fleiss' score is significantly higher: $\kappa = 0.543$. Hence, we conclude that Fleiss' kappa is not a reliable measure of inter-annotator agreement in relation to relatedness scores. Therefore, we decided to use Krippendorff's $\alpha$ instead.

Krippendorff's $\alpha$ (Krippendorff, 1980, 2013) is a coefficient appropriate for measuring inter-annotator agreement of a dataset which is annotated with multiple judges and characterised by different magnitudes of disagreement and missing values. Krippendorff (1980; 2013) offers distance metrics suitable for various scales: binary, nominal, interval, ordinal, and ratio. In ordinal measurement[4] the attributes can be rank-ordered, but

---

[4]Nominal measurement is useless for measuring agreement between relatedness scores ($\alpha = 0.336$ is the identical value as Fleiss' kappa, since all disagreements are considered equal). We also test interval measurement, in which the distance between the attributes does have meaning and an average of an interval variable is computed. The interval score measured for relatedness annotations is quite high $\alpha = 0.785$, but we doubt whether the distance between relatedness scores is meaningful in this case.

distances between them do not have any meaning. Measured with Krippendorff's $\alpha$, there is inter-annotator agreement of $\alpha = 0.780$ for relatedness scores in the Polish evaluation dataset, which is quite satisfactory as well. Hence, we conclude that our dataset is a reliable resource for purpose of evaluating compositional distributional semantics model of Polish.

## 5 Conclusions

The goal of this paper is to present the procedure of building the Polish evaluation dataset for validation of compositional distributional semantics models. As we aim at building an evaluation dataset which is comparable to the SICK corpus, the general assumptions of our procedure correspond to the design principles of the SICK corpus. However, the procedure of building the SICK corpus cannot be adapted without modifications. First, the Polish seed-sentences have to be written based on the images which are selected from 8K ImageFlickr dataset and split into thematic groups, since usable datasets are not publicly available. Second, as the process of transforming sentences seems to be language-specific, the linguistic transformation rules appropriate for Polish have to be defined from scratch. Third, the process of arranging Polish sentences into pairs is defined anew taking into account the data characteristic and bi-directional entailment annotations. The discrepancies relative to the SICK procedure also concern the annotation process itself. Since an entailment relation between two sentences must not be symmetric, each sentence pair is annotated for entailment in both directions. Furthermore, we introduce an element of human verification of correctness of automatically transformed sentences and some additional post-corrections.

The presented procedure results in building the Polish test corpus of relatively high quality. There are inter-annotator agreement on entailment labels of $\kappa = 0.732$ measured with Fleiss' kappa and inter-annotator agreement on relatedness scores of $\alpha = 0.78$ measured with Krippendorff's alpha. The evaluation dataset will be made publicly available upon publication of this paper.

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
