# Peer review of "Polish evaluation dataset for compositional distributional semantics models"

_ACL 2017 — decision unknown_

[Official Review · Reviewer 1 · rating 4 · confidence 4]
soundness 3 · originality 3 · clarity 4 · impact 3 · substance 3 · appropriateness 5 · meaningful comparison 3 · presentation format Poster

- Strengths:

1) This paper proposed a semi-automated framework (human generation -> auto
expansion -> human post-editing) to construct a compositional
semantic similarity evaluation data set.

2) The proposed framework is used to create a Polish compositional semantic
similarity evaluation data set which is useful for future work in developing
Polish compositional semantic models.

- Weaknesses:

1) The proposed framework has only been tested on one language. It is not clear
whether the framework is portable to other languages. For example, the proposed
framework relies on a dependency parser which may not be available in some
languages or in poor performance in some other languages.

2) The number of sentence pairs edited by leader judges is not reported so the
correctness and efficiency of the automatic expansion framework can not be
evaluated. The fact that more than 3% (369 out of 10k) of the post-edited pairs
need further post-editing is worrying. 

3) There are quite a number of grammatical mistakes. Here are some examples but
not the complete and exhaustive list:

line 210, 212, 213: "on a displayed image/picture" -> "in a displayed
image/picture"

line 428: "Similarly as in" -> "Similar to"

A proofread pass on the paper is needed.

- General Discussion: